# Are violence, harmful alcohol/substance use and poor mental health associated with increased genital inflammation?: A longitudinal cohort study with HIV-negative female sex workers in Nairobi, Kenya

Tara S. Beattie[1], James Pollock[2]*, Rhoda Kabuti[3], Tanya Abramsky[1],
Mary Kung'u[3], Hellen Babu[3], The Maisha Fiti Study Champions[3¶], Sanja Huibner[4],
Suji Udayakumar[4], Chrispo Nyamweya[3], Monica Okumu[3], Anne Mahero[3],
Alicja Beksinska[1], Mamtuti Panneh[1], Pauline Ngurukiri[3], Erastus Irungu[3],
Wendy Adhiambo[3], Peter Muthoga[3], Janet Seeley[1], Helen Weiss[5], Rupert Kaul[4‡],
Joshua Kimani[3‡]

1 Department of Global Health and Development, London School of Hygiene and Tropical Medicine, London, United Kingdom, 2 Department of Immunology, University of Toronto, Toronto, Canada, 3 Partners for Health and Development in Africa, Nairobi, Kenya, 4 Department of Medicine, University of Toronto, Toronto, Canada, 5 MRC International Statistics and Epidemiology Group, Department of Infectious Disease Epidemiology, London School of Hygiene and Tropical Medicine, London, United Kingdom

ⓔ These authors contributed equally to this work.
‡ RK and JK also contributed equally to this work.
¶ Membership of The Maisha Fiti Study Champions[3] is provided in the Acknowledgments.
* james.pollock@mail.utoronto.ca

## Abstract

Violence, alcohol use, substance use and poor mental health have been linked with increased HIV acquisition risk, and genital inflammation enhances HIV susceptibility. We examined whether past 6 month experience of these exposures was associated with increased genital inflammation, thereby providing a biological link between these exposures and HIV acquisition risk. The Maisha Fiti study was a longitudinal mixed-methods study of female sex workers in Nairobi, Kenya. Behavioural-biological surveys were conducted at baseline (June-December 2019) and endline (June 2020-March 2021). Analyses were restricted to HIV-negative women (n = 746). Women with raised levels of at least 5 of 9 genital inflammatory cytokines were defined as having genital inflammation. Multivariable logistic regression models were used to estimate (i) baseline associations between genital inflammation and violence, harmful alcohol/substance use, and poor mental health, and (ii) longitudinal associations between these exposures at different survey rounds, and genital inflammation at follow-up. Inflammation data was available for 711 of 746 (95.3%) women at baseline; 351 (50.1%) had genital inflammation, as did 247 (46.7%) at follow-up. At baseline, 67.8% of women had experienced physical and/or sexual violence in the past 6 months, 33.9% had harmful alcohol use, 26.4% had harmful substance use, 25.5% had moderate/severe depression/anxiety, and 13.9% had post-traumatic stress disorder. In adjusted analyses, there was no evidence that these exposures were associated cross-sectionally or

**Data Availability Statement:** The data are not publicly available due to the need to protect participant confidentiality and safety. Our commitment to participant confidentiality and safety is detailed in our participant information sheet and consent forms, as approved by three ethics committees: Kenyatta National Hospital (P778/11/2018), The London School of Hygiene and Tropical Medicine (16229) and the University of Toronto (37046). Requests to access data should be directed to: researchdatamanagement@lshtm.ac.uk, citing item 3643 - The Maisha Fiti Study.

**Funding:** The Maisha Fiti Study is funded by the Medical Research Council MRC and the UK Department of International Development (DFID) (MR/R023182/1) under the MRC/DFID Concordat agreement. HW is supported by the MRC and the DFID under the MRC/DFID Concordat agreement and is also part of the EDCTP2 programme supported by the European Union. RK is supported by the Canadian Institute of Health Research (CIHR) grant #PJT-180629 and #PJT-156123. The funders had no role in study design, data collection and analysis, decision to publish, or preparation of the manuscript.

**Competing interests:** The authors have declared that no competing interests exist.

longitudinally with genital inflammation. We report no associations between past 6 month experience of violence, harmful alcohol/substance use, or poor mental health, and immune parameters previously associated with HIV risk. This suggests that the well-described epidemiological associations between these exposures and HIV acquisition do not appear to be mediated by genital immune changes, or that any such changes are relatively short-lived. High prevalences of these exposures suggest an urgent need for sex-worker specific violence, alcohol/substance use and mental health interventions.

## Introduction

Female sex workers (FSWs) are at increased relative risk of violence experience, harmful alcohol and substance use and mental health problems, compared with women in the general population [1–4]. These factors have multiple harmful social, economic and health consequences including homicide, loss of earnings, breakdown of relationships, direct effects of violence such as physical disability, poor mental health and increased HIV/STIs, and alcohol and drug related harms such as liver disease, and premature death [5–7]. Notably, the relative risk of acquiring HIV is 30 times greater among FSWs compared with women of reproductive age from the general population [1, 8–11]. However, the associations between these exposures—in particular intimate partner sexual violence—and HIV risk may not be entirely explained by behavioural and occupational risk pathways [10].

In recent years, research has demonstrated that genital inflammation increases the risk of HIV acquisition [12]. At the molecular level, inflammation in the female genital tract increases susceptibility to HIV by recruiting and activating local HIV target cells, by reducing epithelial barrier integrity, and by directly promoting HIV replication [4, 13, 14]. A cohort study among women in South Africa found that genital inflammation (based on cytokine levels) was strongly associated with subsequent HIV seroconversion [15]: the cause of this genital inflammation was often not clear. Specifically, inflammation was often apparent without vaginal discharge, genital ulceration, HSV-2 infection or other STIs, and was independent of contraceptive choice, intravaginal substance use, presence of a stable partner, parity, and reported sex acts per month. Importantly, behavioural risk factors were not measured in this study. In contrast, cohort studies with individuals who are highly HIV-Exposed but Seronegative (HESN) suggest that a quiescent genital immune milieu, with lower levels of inflammatory cytokines, may protect against HIV acquisition [16–20].

There are physiological and immunological reasons to suggest that violence experience could increase HIV susceptibility by increasing genital inflammation, potentially through a mental health pathway. A small cross-sectional study with 38 women from the US found systemic and mucosal immune dysregulation in women exposed to sexual violence in the previous 12 weeks [21]. However, to our knowledge, there have been no large-scale or longitudinal epidemiological studies to investigate this. Violence exposure can result in psychological morbidities including depression, anxiety, post-traumatic stress disorder (PTSD) and suicide [22], and poor mental health can be a precursor as well as a consequence of violence [23]. Women who have experienced intimate partner violence (IPV) and meet clinical criteria for PTSD can have elevated CRP levels, a biomarker of systemic inflammation [24]. The hypothalamic pituitary adrenal (HPA) axis produces cortisol during acute stress, and levels of this anti-inflammatory hormone can become dysregulated in individuals with PTSD, resulting in systemic hyper-immune activation [25]. If this systemic inflammation translates into increased genital inflammation, this could enhance HIV susceptibility.

Similarly, heavy alcohol use is independently associated with HIV incidence [26], but the mechanisms remain unclear. Harmful drinking may increase the risk of HIV infection by promoting the likelihood of condomless sex [27], and it has been found that any receptive penile-vaginal sex increases genital inflammation [28]. Alcohol can also be immunosuppressive or immune activating, depending on the frequency and type of use [29]. Intoxicating doses of alcohol are generally immunosuppressive whereas chronic alcohol consumption can cause chronic immune activation [29]. Broad immune dysregulation has been observed in people that smoke tobacco [30] and people with cocaine use disorder [31], while animal studies have described the immunosuppressive role of cannabidiol (CBD), the main cannabinoid found in cannabis [32]. Again, if this systemic inflammation translates into increased genital inflammation, this would increase HIV susceptibility.

Violence experience, harmful alcohol or substance use, and poor mental health can be closely correlated, with bi-directional associations [23, 33, 34]. For example, several studies have found that harmful drinking is associated with incident IPV, and conversely, experience of violence is predictive of increased alcohol or substance use as a coping mechanism [23]. Understanding these pathways will help elucidate potential immune-biological mechanisms through which violence experience, harmful alcohol/substance use and/or poor mental health may be acting on the immune system. Any inflammatory effect of these factors, through local genital inflammation or systemic inflammation that translates to the genital tract, could have an impact on the HIV pandemic in key populations that are disproportionately affected by these factors.

In this paper, we examine cross-sectional and longitudinal associations between genital inflammation and exposure to violence experience in the past 6 months, harmful alcohol/substance use, and mental health problems (depression/anxiety; PTSD) among a cohort of HIV-negative FSWs in Nairobi, Kenya.

## Methods

### Study setting and design

The Maisha Fiti study was a mixed-methods longitudinal cohort study which aimed to examine the biological impact of violence and harmful drinking on inflammatory biomarkers in the blood and the genital tract among a cohort of FSWs in Nairobi, Kenya. Kenya has one of the largest numbers of people living with HIV globally. The adult HIV prevalence is estimated to be 4.0% and is higher in women (5.4%) compared with men (2.6%) [35]. Nairobi is the largest city in Kenya, with a population of ~4.4 million people [36]. Nairobi county has an estimated 2032 locations where approximately 39,600 women sell sex [37], and in 2023 had the highest HIV incidence in Kenya (1999 new infections), followed by Kisumu county (1730 new infections) and Homa Bay county (1495 new infections) [38]. Violence and mental health issues are also commonly experienced by FSWs in Nairobi [39]. Around 73.2% (29,000) of Nairobi FSWs are currently served by seven Sex Worker Outreach Programme (SWOP) clinics which provide peer education and outreach, comprehensive clinical services, including HIV testing and treatment, and condom distribution [40].

A prospective cohort study design was used to be able to assess the causal relationship between violence, mental health issues, alcohol/substance use, and genital inflammation among HIV-negative participants, and to study how exposure to violence changes over time. We did not include HIV incidence as an end-point as we did not have sufficient study power to examine this. The initial study design comprised a behavioural-biological survey completed at two time points—a baseline survey (June-December 2019), and a follow-up survey completed 6–12 months later (January-June 2020). However, the follow-up survey was interrupted

by the onset of the COVID-19 pandemic, resulting in only a small sub-sample of women completing a second survey (January-March 2020) (mid-line). Therefore, the decision was taken to conduct a further follow-up survey (endline) (June 2020-March 2021). Analyses for this paper use data from baseline and endline surveys.

## Sampling

The study was designed in consultation with the FSW community in Nairobi, as well as with peer educators and staff working at the seven SWOP clinics. Participants were randomly selected (using unique enrolment numbers) from all SWOP clinic attendees who had accessed SWOP services in the past 12 months, were aged 18–45 years, and did not have an underlying chronic illness (other than HIV) that was likely to alter host immunology. 10,292 of 29,000 FSWs met these inclusion criteria and were included in the sampling frame. Additional exclusion criteria (assessed during study enrolment) were current pregnancy or breastfeeding.

Sample size calculations have been described previously [41]. To detect a 10% difference in the proportion of women who have genital inflammation (25% vs. 15%) at 90% power, 750 HIV-negative women would be required, assuming 2:1 exposure of recent sexual or physical violence. As HIV prevalence among FSWs attending SWOP clinics in Nairobi is approximately 25%, the target sample size was 1000 FSWs. Oversampling was used to allow for non-eligibility and non-response—thus 1200 women were randomly selected for participation. Numbers selected per clinic were proportional to clinic size and all peer educators (n = 300 for the programme) sensitized and encouraged to create a demand for the study. Women aged <25 years were oversampled to allow sufficient power for analyses stratified by age.

## Recruitment plan

Selected women were telephoned and invited to attend the study clinic, where they were screened for eligibility and received detailed study information verbally in English or Swahili. Consenting women provided written informed consent, with consent obtained at each study visit.

Considerations were made to minimize the risk of harm occurring as a result of participating in this study. For example, the study was locally advertised as a women's health study, not a violence study. The study clinic was located in downtown Nairobi on the fifth floor of an office block, away from where women reside, and minimising the chance of women being identified as a sex worker by their local communities. Women disclosing experiences of recent violence, mental health problems or suicidal behaviours were referred to a trained counsellor within the study team. All women who tested positive for HIV were counselled and referred for HIV care at their chosen SWOP clinic. All women who tested positive for STIs were offered treatment free of charge.

## Ethics statement

All participants provided written formal consent. The Maisha Fiti study was approved by the Kenyatta National Hospital Ethics and Research Committee (KNH ERC P778/11/2018), the London School of Hygiene and Tropical Medicine (LSHTM) Ethics Committee (Approval number: 16229) and the University of Toronto ethics committee (Approval number: 37046).

## Behavioural-biological survey and exposure variables

The behavioural survey collected information on socio-demographics, sexual behaviours/partners, sex work characteristics, STI/HIV testing and treatment, Adverse Childhood Experiences

(ACEs), experiences of violence, mental health, and alcohol and substance use. Interviews were conducted face-to-face in English or Swahili using paper questionnaires administered by trained study team members. At endline, a small minority of participants (n = 23) had left Nairobi and their follow-up survey were conducted by telephone; these data were not included in the current analyses due to the lack of accompanying biological data.

Details of exposure and outcome variables are provided in S1a Table.

The WHO Violence Against Women 13-item questionnaire [42], was used to measure women's lifetime and past 6 months experiences of physical, sexual and emotional Intimate Partner Violence (IPV). We also asked about forced vaginal sex in the past 7 days and included this in the sexual violence exposure measure. The same set of questions were also asked in relation to violent acts perpetrated by non-IPs (e.g. clients, police, strangers, etc). We also asked about experiences of gang rape (ever and past 6 months) and included this in the non-IP sexual violence exposure measure.

We defined two baseline violence exposure variables. Physical and sexual violence (by any perpetrator) were considered in one composite exposure variable due to their high levels of co-occurrence [43]. Since we posited sexual violence as the type of violence likely to be the most proximal and direct influence on genital inflammation, *past 6 months experience of physical and/or sexual violence* was coded as a categorical variable—none, physical only, any sexual (with or without physical). Emotional abuse was considered separately due to hypothesised differences in pathways through which it may affect genital inflammation, and less academic consensus surrounding its measurement [44]. In line with new methodological approaches, we defined *past 6 months experience of emotional violence* as a 3-level ordered categorical variable according to the breadth and frequency of emotionally abusive acts experienced (Low/none, Moderate intensity, High intensity) [45, 46]. We also created two 'trajectory of violence exposure' variables using data from baseline and follow-up: *trajectory of sexual violence experience* and *trajectory of moderate/high intensity emotional violence* (Never, Baseline only, Follow-up with or without baseline).

We defined two baseline mental health exposure variables. Depression was measured using the Patient Health Questionnaire (PHQ-9) (score none 0–4; mild 5–9; moderate-severe 10+), and anxiety was assessed with the Generalised Anxiety Disorder (GAD-7) tool (score none 0–4; mild 5–9; moderate-severe 10+) [47–51]. Due to their high level of co-occurrence [41], we created a combined ordered categorical exposure variable, *anxiety and/or depression* (based on women's highest score for depression or anxiety—none; mild; moderate/severe). Post Traumatic Stress Disorder (PTSD) was measured using the Harvard Trauma Questionnaire (HTQ-17), with a score of > = 2.0 classed as positive for *PTSD* [50]. We also created two 'trajectory of mental health exposure' variables: *trajectory of moderate/severe anxiety and/or depression*; and *trajectory of PTSD experience* (Never, Baseline only, Follow-up with or without baseline).

The WHO ASSIST (Alcohol, Smoking and Substance Involvement Screening Test) tool was used to measure harmful alcohol and other substance use in the past three months [51]. We defined two baseline alcohol and substance use exposure variables: *harmful alcohol use* (Low risk score <11; Moderate risk score > = 11; High risk score >27) and *other harmful substance use* (Low risk score <5; Moderate risk score > = 5; High risk score >27). We also generated two 'trajectory of alcohol/substance use exposure' variables: *trajectory of moderate risk alcohol use* and *trajectory of moderate risk substance use* (Never; Baseline only; Follow-up with or without baseline).

## Laboratory methods

Laboratory methods to test for pregnancy, STIs and HIV are described in S1b Table.

Genital inflammation was assessed through cytokine and chemokine immunoassays. Genital secretions were collected in a SoftCup inserted into the vagina for forty-five to ninety minutes and cryopreserved at -80C. Samples were shipped to Toronto (December 2020 –June 2021) for soluble immune factor measurement. Levels of 9 genital mucosal inflammatory cytokines and chemokines (macrophage inflammatory protein [MIP]-1α, MIP-1β, interferon-γ inducible protein [IP]-10, interleukin [IL]-8, IL-1α, IL-1β, IL-6, monocyte chemoattractant protein [MCP]-1 and tumour necrosis factor [TNF]), were quantified using a highly sensitive electrochemiluminescent multiplex immunoassay (MesoScale Discovery, MSD) as previously described [13, 52]. Samples from the same woman collected at different time-points were run on the same plate to reduce the impact of plate to plate variability on individual women's results.

## Conceptual framework

We developed a conceptual framework linking the exposures (violence experience, harmful alcohol/substance use and mental health) to the outcome (genital inflammation), using a structural determinants and eco-social perspective [53–55] (Fig 1). These theories describe how structural and social determinants or 'exposures beyond an individual's control' (e.g. violence

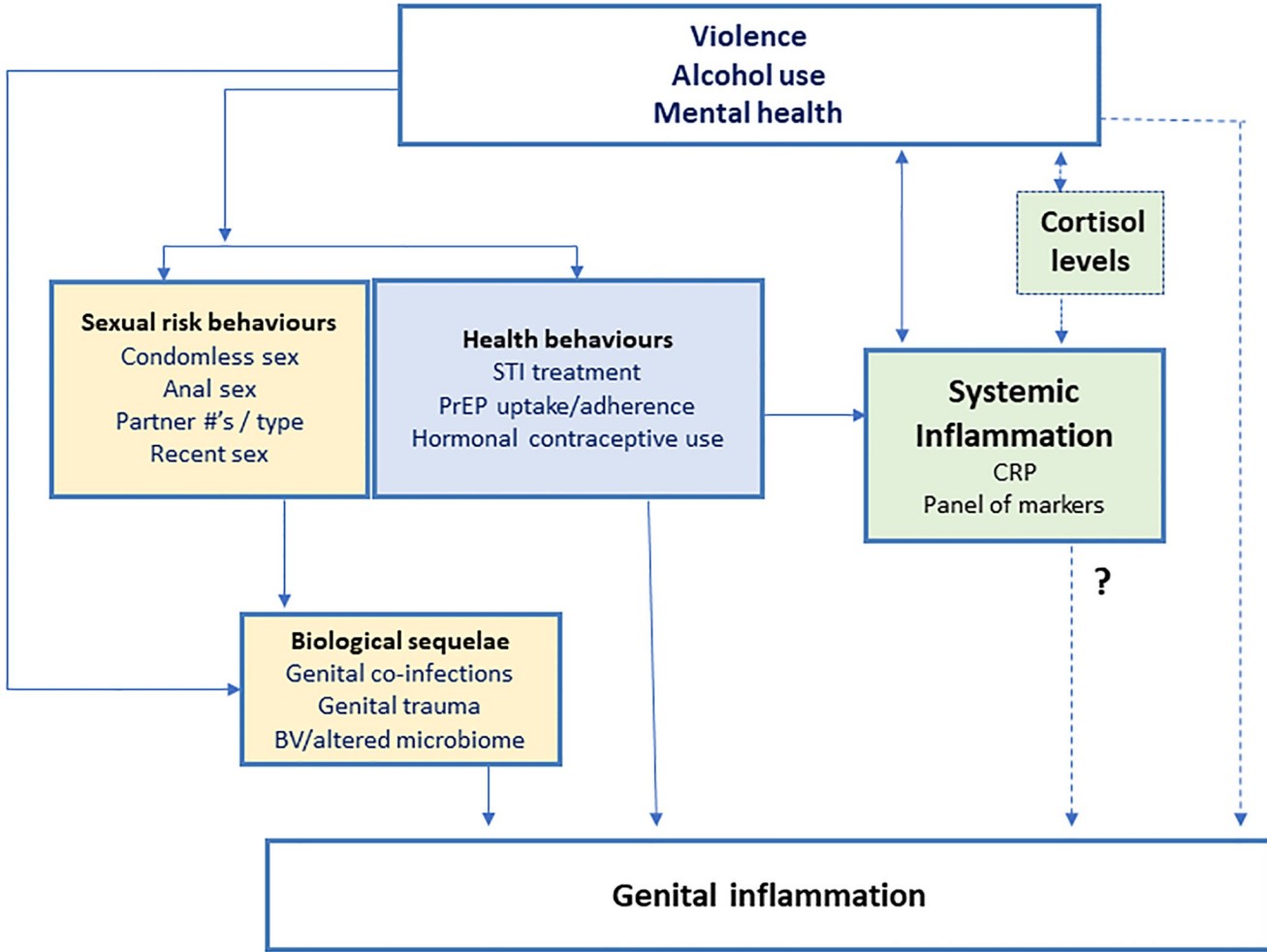

**Fig 1. Conceptual framework exploring potential associations of genital inflammation.**

experience) may become biologically integrated (e.g. genital inflammation) through direct (e.g. rape) and/or indirect (e.g. cortisol induced dysregulation of the HPA) pathways. The framework was informed through global reviews of the current violence against women, harmful alcohol/substance use, mental health and HIV literatures, which identified three potential pathways through which these exposures may influence the outcome (genital inflammation): 1) through increased sexual risk behaviours/genital trauma; 2) through impacts on health behaviours (such as HIV/STI prevention/treatment uptake and adherence, contraceptive choices); and 3) through activation of systemic immune responses. The literature reviews also identified potential confounders and effect modifiers of the exposure/outcome associations. Details on variables considered as potential confounders or effect modifiers in our conceptual framework are provided in S1 Table.

## Statistical analyses

Data were double-entered and statistical analyses were conducted in STATA 16.1 (Stata Inc., College Station, Texas, USA). As women <25 years old were deliberately over-sampled, data were weighted for age during analysis.

We used a ***binary measure of genital inflammation*** in our analysis. While genital inflammation can be defined using several parameters, such as immune cell activation or the elevation of an individual proinflammatory cytokine, we chose this composite score of proinflammatory soluble immune factors as it has been previously demonstrated to be associated with subsequent HIV seroconversion in a study of South African women by Masson et al. [15]. Women with raised levels of at least 5 out of 9 inflammatory cytokines were defined as having genital inflammation. While Masson et al. defined raised as being above the 75[th] percentile for the respective cytokine, we chose to use a median threshold (based on the baseline distribution of values). Our reasons were two-fold: firstly, we hypothesised that levels of inflammation would be higher among FSW than among the Masson study population, and so a focus on the top quartile would exclude many women with high absolute cytokine levels; secondly, use of the median cut-off increased the statistical power of our analyses.

In cases where women did not have complete data for all of the cytokines/chemokines, the binary measure of genital inflammation was still assigned if the available data were sufficient to allow definitive classification of inflammation status (i.e. if there were no permutations through which the missing cytokine data could change the classification). Where, however, the missing data precluded certainty as to inflammation status, women were coded as 'missing' for the genital inflammation outcome.

The analysis comprised a cross-sectional and a longitudinal component and was restricted to HIV- negative women. The cross-sectional analysis explored associations between baseline exposure measures and the baseline genital inflammation outcome. The longitudinal analysis explored associations between the 'trajectory of exposure' measures and the follow-up genital inflammation outcome.

Crude odds ratios (aOR) of association between each of the exposure variables and the relevant genital inflammation outcome were estimated using logistic regression. Accompanying p-values were obtained using a joint hypothesis test via the adjusted Wald test.

Adjusted odds ratios were estimated using multivariable logistic regression. Each of the exposure variables were explored in separate models which included covariates identified *a priori* in the conceptual framework as potential confounders of the exposure/outcome associations. These included age, current intimate partner, education, smoking, female genital mutilation (FGM) and vaginal douching. These are all variables known to be (or plausibly) associated with violence/alcohol or substance use/mental health and/or genital inflammation, but that are

not on the causal pathway between exposures and outcome [56]. SWOP clinic was included as a fixed effect to adjust for clustering by clinic. The same covariates were used in all models to allow comparisons between results. We also tested for interaction between each of the exposure variables and woman's age (using the adjusted Wald test).

Since we had modified the method of defining inflammation compared with the original paper by *Masson et al*, we also conducted a sensitivity analysis, in which we re-ran the above models using the Masson definition of genital inflammation [15].

## Results

### Response rates and completeness of outcome data

Of 1200 women who were invited to participate in the study, 1039 were eligible (not pregnant or breastfeeding and met the other study eligibility criteria). 1003 of 1039 (96.5%) consented to participate and completed the baseline survey. Of these, 746 were HIV negative, with 711 (95.3%) having sufficient baseline cytokine data to generate the binary genital inflammation outcome and thus be included in the cross-sectional analysis.

Of the 877 women who completed the follow-up survey (87.4% follow-up), 637 were HIV-negative. There were no HIV seroconversions during the study. Of these 637 participants, 526 (82.6%) had sufficient follow-up cytokine data to generate the genital inflammation outcome and were included in the longitudinal analysis.

### Baseline characteristics

Baseline descriptive data for women included in the baseline cross-sectional analysis are presented in Table 1. The weighted mean age was 32.4 years (SD 7.1). Most women (90.8%) were Catholic or Protestant. 17.0% of respondents had no/below primary education, with a similar number (16.1%) reporting low literacy. A further 51.5% had completed primary school only.

Most women (79.2%) had been married at some point in their lives. However, at the time of the survey, just two-thirds of women (63.6%) had a current intimate partner, and few (7.0%) reported cohabiting with their partner. Most (78.9%) had children living with them. Almost half (44.4%) reported having an income other than sex work, and one-third (32.7%) reported that they and/or other family members had missed a meal in the last week due to financial constraints.

Data on other factors potentially related to genital inflammation are presented in Table 1. Briefly, most women had had sex in the past week, and the median number of clients in the past week was 4 (IQR 1–6). Three-quarters (75.4%) reported using a condom at last vaginal sex. Prevalence of bacterial STIs was relatively low (Chlamydia trachomatis 7.1%, Neisseria gonorrhoeae 2.7%, Syphilis 1.7%), 2.6% had Trichomonas vaginalis, 17.1% tested positive for BV, and 47.6% for HSV-2. A quarter of women (24.8%) reported being currently on PrEP, and 3.5% were currently taking PEP. Half (49.3%) were currently using long-acting reversible contraception. A small number of women (5.8%) reported FGM. Vaginal douching was commonly practised, with 59.3% reporting intravaginal washing in the past 30 days, most using water only.

### Prevalence of genital inflammation

Half of women (50.1%) were classified as having genital inflammation at baseline (95% CI 46.4%–53.7%). Prevalence of genital inflammation was not uniform across all sub-groups. Genital inflammation was increased among those with an intimate partner compared to single women, those with a current STI (specifically gonorrhoea or chlamydia), and those with an

**Table 1. Baseline characteristics and associated prevalence of genital inflammation, among HIV-negative participants with available baseline genital inflammation data.**

| Baseline characteristic | | n (weighted %) N = 711 | Weighted % with genital inflammation at baseline |
|---|---|---|---|
| **Overall** | | **711** | **50.1% (351/711)** |
| **Demographics** | | | |
| Age (years) | 18–24 | 188 (15.1%) | 44.7% |
| | 25–34 | 272 (44.2%) | 52.9% |
| | 35–45 | 251 (40.8%) | 49.0% |
| Religion | Catholic | 270 (37.4%) | 46.2% |
| | Protestant | 371 (53.4%) | 54.6% |
| | Muslim | 34 (4.9%) | 50.0% |
| | Other/none | 34 (4.3%) | 28.3% |
| Education | No/incomplete primary | 112 (17.0%) | 46.7% |
| | Completed primary only | 367 (51.5%) | 52.1% |
| | Completed secondary/higher | 232 (31.5%) | 48.6% |
| Literate | No/partly | 106 (16.1%) | 57.1% |
| | Yes | 605 (83.9%) | 48.8% |
| Ever married | No | 170 (20.8%) | 51.8% |
| | Yes | 541 (79.2%) | 49.7% |
| Has a current intimate partner (non-paying) | No | 251 (36.4%) | 43.9% |
| | Yes | 460 (63.6%) | 53.6% |
| Number of children living with respondent | None | 159 (21.2%) | 46.8% |
| | 1–2 | 412 (58.7%) | 50.2% |
| | 3+ | 139 (20.2%) | 53.4% |
| SES tertile | Lower | 236 (32.2%) | 50.0% |
| | Middle | 225 (31.7%) | 52.5% |
| | Upper | 250 (36.1%) | 48.1% |
| Income other than sex work | No | 399 (55.6%) | 48.3% |
| | Yes | 312 (44.4%) | 52.4% |
| Respondent and/or family skipped meal in past 7 days | No | 486 (67.3%) | 50.1% |
| | Yes | 223 (32.7%) | 49.5% |
| **Sexual risk behaviours and STI** | | | |
| Timing of last sex | Within past 3 days | 390 (55.6%) | 50.5% |
| | 4–6 days ago | 190 (26.9%) | 49.1% |
| | More than 6 days ago | 130 (17.5%) | 50.5% |
| Condom at last vaginal sex | No | 181 (24.6%) | 55.3% |
| | Yes | 529 (75.4%) | 48.5% |
| Condom at last sex with a client | No | 125 (17.3%) | 50.7% |
| | Yes | 578 (82.7%) | 50.3% |
| Number of clients in past week | <5 | 416 (58.8%) | 50.3% |
| | 5–9 | 192 (26.9%) | 49.4% |
| | 10+ | 97 (14.3%) | 50.3% |
| Current gonorrhoea infection | No | 690 (97.3%) | 49.5% |
| | Yes | 20 (2.7%) | 69.7% |
| Current chlamydia infection | No | 651 (92.9%) | 48.7% |
| | Yes | 58 (7.1%) | 67.1% |
| Current gonorrhoea or chlamydia | No | 635 (90.5%) | 48.3% |
| | Yes | 76 (9.6%) | 67.0% |

*(Continued)*

**Table 1.** (Continued)

| Baseline characteristic | | n (weighted %) N = 711 | Weighted % with genital inflammation at baseline |
|---|---|---|---|
| Current syphilis infection | No | 696 (98.3%) | 50.2% |
| | Yes | 11 (1.7%) | 42.8% |
| Current trichomonas infection | No | 689 (97.4%) | 49.7% |
| | Yes | 19 (2.6%) | 56.2% |
| HSV-2 | No/borderline | 345 (44.6%) | 53.9% |
| | Yes | 358 (55.4%) | 47.6% |
| Bacterial vaginosis (BV) | Normal | 329 (45.7%) | 45.5% |
| | Abnormal Vaginal Flora (AVF) | 257 (37.2%) | 60.5% |
| | BV | 123 (17.1%) | 39.1% |
| **Health-seeking/treatment uptake** | | | |
| Currently taking PrEP | No | 536 (75.2%) | 47.8% |
| | Yes | 172 (24.8%) | 57.4% |
| Missed any PrEP doses in last month | No | 119 (68.6%) | 52.9% |
| | Yes | 52 (31.4%) | 66.4% |
| Currently taking PEP | No | 683 (96.5%) | 50.0% |
| | Yes | 23 (3.5%) | 51.2% |
| Using long-acting reversible contraception | No | 353 (50.7%) | 48.9% |
| | Yes | 358 (49.3%) | 51.3% |
| **Vaginal factors** | | | |
| Surgical procedures to modify vagina | Never | 672 (94.2%) | 50.7% |
| | Ever | 39 (5.8%) | 39.5% |
| Intravaginal washing practices (past 30 days) | None | 288 (40.7%) | 53.2% |
| | Water only | 349 (49.6%) | 48.5% |
| | Something other than water | 71 (9.7%) | 44.6% |
| **Other factors** | | | |
| Schistosomiasis | No | 577 (81.9%) | 49.2% |
| | Yes | 134 (18.1%) | 54.3% |
| Tobacco use (past 3 months) | None | 569 (80.0%) | 53.3% |
| | Some | 142 (20.0%) | 37.3% |

intermediate Nugent score, although not those with BV. It was also higher among women currently taking PrEP, compared to women not on PrEP; however, among current PrEP users, those who reported adhering to PrEP were less likely to have genital inflammation than those who missed doses. Prevalence of genital inflammation did not vary across age-groups.

At follow-up, 46.7% of participants were classified as having genital inflammation (95% CI 42.5%–51.0%). Those with baseline inflammation were more likely to have inflammation at follow-up compared to those without baseline inflammation (58.1% versus 35.1%).

## Violence, harmful alcohol/substance use, and mental health exposures

**Violence.** At baseline, 67.8% of women had experienced physical and/or sexual violence in the past 6 months. Emotional violence (moderate or high intensity) in the past 6 months was also highly prevalent (61.3%).

At baseline, experience of physical violence alone was not associated with genital inflammation. However, women with experience of sexual violence in the past 6 months had reduced odds of genital inflammation compared to women without past 6 month experience of

physical or sexual violence (aOR 0.70; 95% CI 0.50–0.98). This association was not replicated longitudinally—there was no evidence of an association between trajectory of exposure to sexual violence over the course of the study and genital inflammation at follow-up.

Odds of genital inflammation at baseline were lower (not statistically significant) among women experiencing high intensity emotional violence compared to those with no emotional violence (aOR 0.72; 95% CI 0.46–1.11), though no consistent trend was observed in relation to increasing exposure intensity. There was no evidence of an association between trajectory of exposure to moderate/high intensity emotional violence and genital inflammation at follow-up.

**Alcohol and substance use.**   Levels of harmful alcohol use were high at baseline, with a third (33.9%) of women categorised in the moderate or high-risk category. Harmful substance use (other than alcohol or tobacco) was also prevalent, with a quarter of women (26.4%) in the moderate risk category (very few were categorised as high risk).

At baseline, high risk alcohol use was associated with lower odds of genital inflammation compared to low risk alcohol use (crude OR 0.58; 0.36–0.96), though this lost statistical significance after inclusion of smoking in the multivariate model (aOR 0.67; 0.41–1.10; Table 2). Moderate alcohol use showed no association with genital inflammation. No associations were observed between trajectory of exposure to alcohol risk and genital inflammation at follow-up (Table 3).

No evidence of associations were observed between non-alcohol substance use risk level and genital inflammation at baseline, though small numbers in the highest substance use risk category limited power to draw conclusions about potential effects of high-risk substance use. Similarly, no associations were observed between trajectory of exposure to substance use risk and genital inflammation at follow-up.

**Mental health.**   Over half (54.0%) of women were classed as having at least mild symptoms of anxiety and/or depression in the past 2 weeks at baseline, with a quarter (25.5%) in the moderate/severe category. 13.9% screened positive for PTSD in the past month.

There was no evidence of an association between either anxiety/depression or PTSD and genital inflammation at baseline, or longitudinally.

**Interactions between exposure variables and age.**   No consistent patterns of interaction were observed between any of the exposure variables and age.

**Sensitivity analysis.**   Findings using the Masson definition of genital inflammation were similar to those using our modified definition (S2a and S2b Table), with no significant associations seen longitudinally between the violence, mental health and alcohol/substance use exposure variables and genital inflammation.

## Discussion

Among 711 HIV-negative FSWs in Nairobi, Kenya, we found no evidence of associations between the presence of genital inflammation, defined as increased levels of ≥5 of 9 vaginal inflammatory/chemotactic immune factors, and either any reported sexual/physical violence experience during the past 6 months, harmful alcohol or substance use, or anxiety/depression or PTSD. While the cross-sectional analyses demonstrated that any sexual violence in the past 6 months was weakly associated with reduced odds of genital inflammation, the confidence intervals were wide and this association was not seen in longitudinal analysis, suggesting that this may not reflect a true biologic association. A sensitivity analysis using a previously cited definition of genital inflammation found similar results.

Although we did not find evidence of any associations with increased inflammation cross-sectionally or longitudinally, we are currently examining associations between these exposure

**Table 2. Odds ratios of association between baseline exposure variables and baseline genital inflammation outcome.**

| Baseline exposures | | n (weighted %) N = 711 | Weighted % with genital inflammation at baseline | OR (95% CI) [Adjusted Wald test p-value] | aOR* (95%CI) [Adjusted Wald test p-value] |
|---|---|---|---|---|---|
| **Violence exposures (past 6 months by any perpetrator)** | | | | | |
| Experience of physical and sexual violence | None | 236 (32.2%) | 55.0% | - | - |
| | Physical only | 106 (14.5%) | 52.5% | 0.90 (0.57–1.43) | 0.90 (0.56–1.44) |
| | Sexual (with or without physical) | 369 (53.3%) | 46.4% | 0.71 (0.51–0.98) | 0.70 (0.50–0.98) |
| | | | | [p = 0.104] | [p = 0.102] |
| Emotional violence | None | 282 (38.7%) | 49.6% | - | - |
| | Moderate | 302 (43.5%) | 53.8% | 1.18 (0.85–1.63) | 1.13 (0.81–1.58) |
| | Severe | 127 (17.8%) | 42.3% | 0.74 (0.49–1.14) | 0.72 (0.46–1.11) |
| | | | | [p = 0.093] | [p = 0.118] |
| **Alcohol and substance abuse (past 3 months)** | | | | | |
| Alcohol ASSIST risk level | Low (0–10) | 466 (66.1%) | 51.2% | - | - |
| | Moderate (11–26) | 163 (22.7%) | 52.0% | 1.03 (0.72–1.47) | 1.06 (0.72–1.56) |
| | High (27+) | 78 (11.2%) | 38.0% | 0.58 (0.36–0.96) | 0.67 (0.41–1.10) |
| | | | | [p = 0.087] | [p = 0.239] |
| Other substance use ASSIST risk level (not alcohol or tobacco) | Low | 519 (73.6%) | 51.3% | - | - |
| | Moderate/High | 192 (26.4%) | 46.6% | 0.83 (0.59–1.15) | 0.95 (0.66–1.38) |
| | | | | [p = 0.265] | [p = 0.801] |
| **Mental health exposures** | | | | | |
| Anxiety and/or depression (past 2 weeks) | Low | 336 (46.0%) | 47.9% | - | - |
| | Mild | 202 (28.5%) | 54.3% | 1.29 (0.91–1.83) | 1.25 (0.86–1.81) |
| | Moderate/severe | 173 (25.5%) | 49.4% | 1.06 (0.74–1.53) | 1.15 (0.78–1.69) |
| | | | | [p = 0.349] | [p = 0.477] |
| PTSD (past month) | No | 609 (86.1%) | 49.4% | - | - |
| | Yes | 95 (13.9%) | 53.5% | 1.18 (0.77–1.81) | 1.28 (0.82–1.99) |
| | | | | [p = 0.447] | [p = 0.277] |

*Adjusted for age, education, current intimate partner, FGM, intravaginal washing practices, smoking and SWOP clinic

variables and other factors along the postulated causal pathway including (i) cortisol levels and (ii) systemic inflammation. We will also be investigating associations between cortisol levels and systemic inflammation, and between systemic inflammation and genital inflammation, to better understand potential physiological impacts of these exposure variables and how these relate to HIV acquisition risk. Indeed, a study with FSWs in Mombasa, Kenya, found that physical, sexual or emotional violence in the past 12 months was significantly associated with elevated cortisol levels—one of the factors on our hypothesised causal pathway—but they found no associations between violence and two measures of systemic inflammation (C-reactive protein; IL-6) [57]. Another study with 38 women in the U.S. found associations between sexual violence in the past 12 weeks and individual systemic and genital inflammatory markers [21]. Specifically, they found sexual violence in this time frame to be associated with increased vaginal levels of IL-1α, and with decreased levels of vaginal MCP-1 and MIP-3α. The

**Table 3. Odds ratios of association between trajectory of exposure variables and genital inflammation at follow-up.**

| | n (weighted %) N = 526 | Weighted % with genital inflammation at baseline | OR (95% CI) [Adjusted Wald test p-value] | aOR* (95%CI) [Adjusted Wald test p-value] |
|---|---|---|---|---|
| **Sexual violence (with or without physical violence)** | | | | |
| Never | 225 (41.5%) | 46.5% | - | - |
| Baseline only | 198 (38.8%) | 48.0% | 1.06 (0.73–1.56) | 1.06 (0.72–1.57) |
| Follow-up (with or without baseline) | 103 (19.7%) | 44.7% | 0.93 (0.59–1.48) | 0.92 (0.57–1.48) |
| | | | [p = 0.854] | [p = 0.835] |
| **Moderate/severe emotional violence** | | | | |
| Never | 168 (31.3%) | 46.3% | - | - |
| Baseline only | 198 (37.7%) | 47.7% | 1.06 (0.70–1.59) | 1.01 (0.66–1.55) |
| Follow-up (with or without baseline) | 160 (31.1%) | 46.0% | 0.98 (0.64–1.52) | 0.93 (0.60–1.44) |
| | | | [p = 0.940] | [p = 0.911] |
| **Moderate/high risk alcohol use** | | | | |
| Never | 314 (60.3%) | 47.0% | - | - |
| Baseline only | 131 (25.1%) | 42.2% | 0.82 (0.55–1.24) | 0.81 (0.53–1.23) |
| Follow-up (with or without baseline) | 81 (14.7%) | 53.3% | 1.29 (0.79–2.10) | 1.27 (0.76–1.23) |
| | | | [p = 0.285] | [p = 0.284] |
| **Moderate/high risk substance use** | | | | |
| Never | 331 (64.5%) | 47.3% | - | - |
| Baseline only | 59 (10.8%) | 50.5% | 1.13 (0.65–1.97) | 1.10 (0.62–1.95) |
| Follow-up (with or without baseline) | 136 (24.7%) | 43.6% | 0.86 (0.58–1.29) | 0.71 (0.45–1.13) |
| | | | [p = 0.638] | [p = 0.280] |
| **Moderate/severe anxiety and/or depression** | | | | |
| Never | 370 (68.9%) | 46.8% | - | - |
| Baseline only | 104 (20.2%) | 46.5% | 0.99 (0.64–1.52) | 1.05 (0.67–1.66) |
| Follow-up (with or without baseline) | 52 (10.9%) | 47.0% | 1.01 (0.57–1.78) | 1.05 (0.59–1.87) |
| | | | [p = 0.998] | [p = 0.969] |
| **PTSD** | | | | |
| Never | 453 (85.8%) | 46.5% | - | - |
| Baseline only | 63 (12.2%) | 52.6% | 1.28 (0.76–2.16) | 1.41 (0.82–2.43) |
| Follow-up (with or without baseline) | 10 (2.0%) | 22.3% | 0.33 (0.07–1.50) | 0.34 (0.08–1.44) |
| | | | [p = 0.218] | [p = 0.143] |

*Adjusted for age, education, current intimate partner, FGM, intravaginal washing practices, smoking and SWOP clinic

bidirectional associations of violence in that study with different components of our composite inflammation definition may explain why we found no overall association of inflammation with sexual violence in our own analysis, but we felt that use of this composite definition was justified by its prior association with HIV acquisition in a prospective cohort [15] and in order to avoid statistical concerns of multiple comparisons. Furthermore, the timeframe of that analysis (12 weeks) differed to ours (6 months). It will be interesting to explore the (potentially interacting) effects of alcohol, substance use and mental health problems on cortisol levels and systemic inflammation, as well as the more immediate effects of sexual violence on genital inflammation.

Of note, we did find extremely high prevalences of all our hypothesised exposure variables. Regardless of the physiological associations—or lack thereof—with increased genital

inflammation, these high prevalences highlight the urgent need for effective interventions for FSWs and their communities to prevent violence, support victims of violence, bring perpetrators of violence to justice and provide effective, low-cost treatments for common mental health problems and harmful alcohol and substance use.

Key strengths of our study include the large sample size, the probabilistic sample selection, the longitudinal study design which would have enabled us to assess causality if associations had been found, and the use of validated tools to measure violence experience, mental health problems, and harmful alcohol and substance use. The use of biological measures for our outcome 'genital inflammation' is another important strength as biological measures are not subject to reporting bias.

Under-reporting of sensitive topics, including violence experience, mental health problems, and alcohol and illegal substance use is a key study limitation and could have resulted in an under-estimation of these exposure prevalences and consequent reductions in study power to measure associations. Although odds ratios are frequently the measure of effect used in cross-sectional and longitudinal studies, as the outcome was common (50%), the OR will be somewhat greater than the prevalence ratio and caution needs to be made when interpreting the magnitude and thereby 'clinical significance' of any observed associations. Indeed, we caution against the over-interpretation of the association we found cross-sectionally between sexual violence and reduced genital inflammation, especially as this makes little sense biologically. We did aim to include all known confounders from the literature in order to adjust for these in our analyses, while not 'over-adjusting' for factors which might be on the causal pathway between exposures and outcome. However, it is plausible that we missed important confounders. We were missing data on individual genital inflammatory cytokine levels for several individuals; however we were able to still include the majority of these individuals in our analyses as we could infer their genital inflammatory 'status' from the readings we did have for them.

Taken together, this study answers important questions as to the impact of 'upstream' structural and behavioural risk factors on genital inflammation and subsequent HIV risk—namely that we found no associations between past 6 months violence, mental health problems or harmful drinking or substance use and increased genital inflammation among this cohort of HIV-negative FSWs in Nairobi, Kenya. Future studies will examine associations between factors on the hypothesised pathway and the impact of more recent sexual violence to help further elucidate immunological mechanisms behind increased risk of HIV acquisition among FSWs. The decriminalisation of sex work, and interventions to address violence, mental health problems and harmful alcohol and substance use are urgently needed for FSWs in this setting.

## Supporting information

**S1 Table. a**: Exposure, outcome, and contextual variables. **b**: Assessment of pregnancy, STIs and HIV.
(DOCX)

**S2 Table. a**. Results of sensitivity analysis—Odds ratios of association between baseline exposure variables and baseline genital inflammation outcome as defined by Masson et al. (2015) using upper quartile as the threshold to define raised levels of each cytokine. **b**. Results of sensitivity analysis—Odds ratios of association between trajectory of exposure variables and genital inflammation at follow-up, as defined by Masson et al. (2015) using upper quartile as the threshold to define raised levels of each cytokine.
(DOCX)

## Acknowledgments

The authors would like to thank all women who participated in this study. They also thank all the Maisha Fiti study champions (Demtilla Gwala, Daisy Oside, Ruth Kamene, Agnes Watata, Agnes Atieno, Faith Njau, Elizabeth Njeri, Evelyn Orobi & Ibrahim Lwingi) for their educational outreach and community engagement.

## Author Contributions

**Conceptualization:** Tara S. Beattie, Rhoda Kabuti, Janet Seeley, Helen Weiss, Rupert Kaul, Joshua Kimani.

**Data curation:** James Pollock, Tanya Abramsky, Sanja Huibner, Suji Udayakumar, Pauline Ngurukiri.

**Formal analysis:** Tara S. Beattie, James Pollock, Tanya Abramsky.

**Funding acquisition:** Tara S. Beattie, Janet Seeley, Helen Weiss, Rupert Kaul.

**Investigation:** Tara S. Beattie, James Pollock, Rhoda Kabuti, Tanya Abramsky, Mary Kung'u, Hellen Babu, Sanja Huibner, Suji Udayakumar, Chrispo Nyamweya, Monica Okumu, Anne Mahero, Mamtuti Panneh, Erastus Irungu, Wendy Adhiambo, Peter Muthoga, Rupert Kaul.

**Methodology:** Tara S. Beattie, James Pollock, Tanya Abramsky, Sanja Huibner, Suji Udayakumar, Alicja Beksinska, Erastus Irungu, Rupert Kaul.

**Project administration:** Rhoda Kabuti, Hellen Babu, Janet Seeley, Helen Weiss, Joshua Kimani.

**Resources:** Erastus Irungu, Rupert Kaul.

**Supervision:** Tara S. Beattie, Rupert Kaul, Joshua Kimani.

**Validation:** James Pollock, Rhoda Kabuti, Tanya Abramsky, Mary Kung'u, Hellen Babu, Sanja Huibner, Suji Udayakumar, Chrispo Nyamweya, Monica Okumu, Anne Mahero, Pauline Ngurukiri.

**Writing – original draft:** Tara S. Beattie, Tanya Abramsky.

**Writing – review & editing:** Tara S. Beattie, James Pollock, Rhoda Kabuti, Tanya Abramsky, Mary Kung'u, Hellen Babu, Sanja Huibner, Suji Udayakumar, Chrispo Nyamweya, Monica Okumu, Anne Mahero, Alicja Beksinska, Mamtuti Panneh, Pauline Ngurukiri, Wendy Adhiambo, Janet Seeley, Helen Weiss, Rupert Kaul, Joshua Kimani.

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
