## [Decision Letter · Decision Letter 0]

29 May 2024

PGPH-D-24-00503

Are violence, harmful alcohol/substance use and poor mental health associated with increased genital inflammation?: A longitudinal cohort study with HIV-negative Female Sex Workers in Nairobi, Kenya

Dear Dr. Pollock,

Thank you for submitting your manuscript to PLOS Global Public Health. After careful consideration, we feel that it has merit but does not fully meet PLOS Global Public Health’s publication criteria as it currently stands. Therefore, we invite you to submit a revised version of the manuscript that addresses the points raised during the review process.

EDITOR: The manuscript has been reviewed. Please review and address the reviewer's comments and provide a point-by-point response to each comment.

We look forward to receiving your revised manuscript.

Kind regards,

Tanmay Bagade, Ph.D., MS (O&G), MPH, MHM

Academic Editor

Journal Requirements:

Additional Editor Comments (if provided):

Reviewers' comments:

Reviewer's Responses to Questions

**Comments to the Author**

1. Does this manuscript meet PLOS Global Public Health’s publication criteria? Is the manuscript technically sound, and do the data support the conclusions? The manuscript must describe methodologically and ethically rigorous research with conclusions that are appropriately drawn based on the data presented.

Reviewer #1: No

Reviewer #2: Partly

2. Has the statistical analysis been performed appropriately and rigorously?

Reviewer #1: Yes

Reviewer #2: Yes

3. Have the authors made all data underlying the findings in their manuscript fully available (please refer to the Data Availability Statement at the start of the manuscript PDF file)?

Reviewer #1: Yes

Reviewer #2: Yes

4. Is the manuscript presented in an intelligible fashion and written in standard English?

Reviewer #1: Yes

Reviewer #2: Yes

5. Review Comments to the Author

Reviewer #1: Are violence, harmful alcohol/substance use and poor mental health associated with increased genital inflammation?: A longitudinal cohort study with HIV-negative Female Sex Workers in Nairobi, Kenya

This is a very innovative study for public health implication. The analysis was thoroughly done, although the overall justification for the study was not adequately discussed.

First, the topic seems inappropriate, especially just mentioning “genital inflammation” this is non specific of a particular or specific diagnosis or infection. It should be stated what exactly the outcome of study is, for example, genital inflammation associated with HIV seroconversion or infection.

Secondly, the conclusion in the abstract states that ...this HIV acquisition risk is more fully explained by behavioural rather than biological pathways. The researcher should be careful of the use of the word fully. This seems inappropriate, no study or findings of a research fully explains the outcome.

Third; line 105 to 110, the explanation on the relationship between alcohol, condomless sex, genital inflammation, and HIV susceptibility is not clear. Is the researcher trying to attribute drinking and condomless sex to genital inflammation? yes alcohol could impair sexual-decision making to engage in condomless sex, and it can decrease immunity but what is the link between these and genital inflammation?

Line 113, remove “...with bi-directional associations” the relationship between those variables are all not bi-directional. Also your example is inappropriate, why use the words “women who are drunk”...you should rather use articles from past studies to butress your point with correct citation.

Line 142; not appropriate “10,292/29,000” does this means 10,292 divides by 29,000? Be clearer.

Line 146; why 1000 FSWs? what informs the sample size determination?

The justification for this study was not clearly explained. Alcohol use is associated with Sexual violence from previous studies, it is also established that sexual violence increases vulnerability to HIV however, the number of HIV incidence from sexual violence is low and also linking all this variables together was not clearly explained in this study.

The researcher should have a discussion section to explicitly discuss the findings and relate to previous studies done relating this variables for clarity.

Reviewer #2: The comments are provided in a separate sheet which is uploaded online.

Also, I mentioned that the paper can be partly published which means, after reviewing and addressing all provided comments from all reviewers. It would be worth to be published

6. PLOS authors have the option to publish the peer review history of their article (what does this mean?). If published, this will include your full peer review and any attached files.

**Do you want your identity to be public for this peer review?** For information about this choice, including consent withdrawal, please see our Privacy Policy.

Reviewer #1: **Yes: **Queen Esther Adeyemo

Reviewer #2: **Yes: **Marie Grace Sandra Musabwasoni

---

## [Editor Report · Decision Letter 1]

24 Jul 2024

Are violence, harmful alcohol/substance use and poor mental health associated with increased genital inflammation?: A longitudinal cohort study with HIV-negative Female Sex Workers in Nairobi, Kenya

PGPH-D-24-00503R1

Dear Mr. Pollock,

We are pleased to inform you that your manuscript 'Are violence, harmful alcohol/substance use and poor mental health associated with increased genital inflammation?: A longitudinal cohort study with HIV-negative Female Sex Workers in Nairobi, Kenya' has been provisionally accepted for publication in PLOS Global Public Health.

Best regards,

Tanmay Bagade, Ph.D., MS (O&G), MPH, MHM

Academic Editor